# Optimal prior-dependent neural population codes under shared input noise

**Agnieszka Grabska-Barwińska**
Gatsby Computational Neuroscience Unit
University College London
agnieszka@gatsby.ucl.ac.uk

**Jonathan W. Pillow**
Princeton Neuroscience Institute
Department of Psychology
Princeton University
pillow@princeton.edu

## Abstract

The brain uses population codes to form distributed, noise-tolerant representations of sensory and motor variables. Recent work has examined the theoretical optimality of such codes in order to gain insight into the principles governing population codes found in the brain. However, the majority of the population coding literature considers either conditionally independent neurons or neurons with noise governed by a stimulus-independent covariance matrix. Here we analyze population coding under a simple alternative model in which latent "input noise" corrupts the stimulus before it is encoded by the population. This provides a convenient and tractable description for irreducible uncertainty that cannot be overcome by adding neurons, and induces stimulus-dependent correlations that mimic certain aspects of the correlations observed in real populations. We examine prior-dependent, Bayesian optimal coding in such populations using exact analyses of cases in which the posterior is approximately Gaussian. These analyses extend previous results on independent Poisson population codes and yield an analytic expression for squared loss and a tight upper bound for mutual information. We show that, for homogeneous populations that tile the input domain, optimal tuning curve width depends on the prior, the loss function, the resource constraint, and the amount of input noise. This framework provides a practical testbed for examining issues of optimality, noise, correlation, and coding fidelity in realistic neural populations.

## 1 Introduction

A substantial body of work has examined the optimality of neural population codes [1–19]. However, the classical literature has focused mostly on codes with independent Poisson noise, and on Fisher information-based analyses of unbiased decoding. Real neurons, by contrast, exhibit dependencies beyond those induced by the stimulus (i.e., "noise correlations"), and Fisher information does not accurately quantify information when performance is close to threshold [7, 15, 18], or when biased decoding is optimal. Moreover, the classical population codes with independent Poisson noise predict unreasonably good performance with even a small number of neurons. A variety of studies have shown that the information extracted from independently recorded neurons (across trials or even animals) outperforms the animal itself [20, 21]. For example, a population of only 220 Poisson neurons with tuning width of 60 deg (full width at half height) and tuning amplitude of 10 spikes can match the human orientation discrimination threshold of $\approx 1$ deg. (See Supplement S1 for derivation.) Note that even fewer neurons would be required if peak spike counts were higher.

The mismatch between this predicted efficiency and animals' actual behaviour has been attributed to the presence of information-limiting correlations between neurons [22, 23]. However, deviation

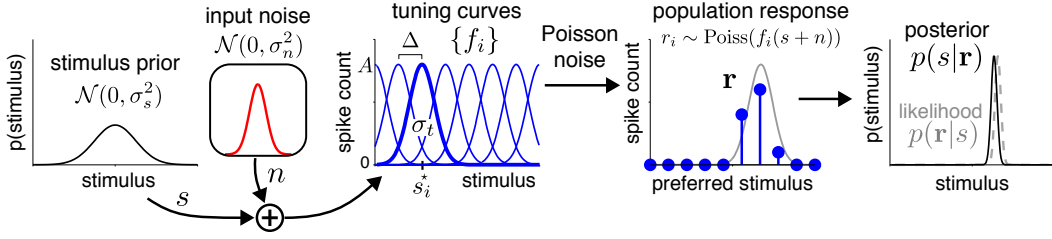

Figure 1: Bayesian formulation of neural population coding with input noise.

from independence renders most analytical treatments infeasible, necessitating the use of numerical methods (Monte Carlo simulations) for quantifying the performance of such codes [7, 15].

Here we examine a family of population codes for which the posterior is Gaussian, which makes it feasible to perform a variety of analytical treatments. In particular, when tuning curves are Gaussian and "tile" the input domain, we obtain codes for which the likelihood is proportional to a Gaussian [2, 16]. Combined with a Gaussian stimulus prior, this results in a Gaussian posterior whose variance depends only on the total spike count. This allows us to derive tractable expressions for neurometric functions such as mean squared error (MSE) and mutual information (MI), and to analyze optimality without resorting to Fisher information, which can be inaccurate for short time windows or small spike counts [7, 15, 18]. Secondly, we extend this framework to incorporate shared "input noise" in the stimulus variable of interest (See Fig. 1). This form of noise differs from many existing models, which assume noise to arise from shared connectivity, but with no direct relationship to the stimulus coding [5, 15, 18, 24] (although see [16, 25] for related approaches).

This paper is organised as follows. In Sec. 2, we describe an idealized Poisson population code with tractable posteriors, and review classical results based on Fisher Information. In Sec. 3, we provide a Bayesian treatment of these codes, deriving expressions for mean squared error (MSE) and mutual information (MI). In Sec. 4, we extend these analyses to a population with input noise. Finally, in Sec. 5 we examine the patterns of pairwise dependencies introduced by input noise.

## 2    Independent Poisson population codes

Consider an idealized population of Poisson neurons that encode a scalar stimulus $s$ with Gaussian-shaped tuning curves. Under this (classical) model, the response vector $\boldsymbol{r} = (r_1, \dots r_N)^\top$ is conditionally Poisson distributed:

$$r_i|s \sim \text{Poiss}(f_i(s)), \qquad p(\boldsymbol{r}|s) = \prod_{i=1}^{N} \tfrac{1}{r_i!} f_i(s)^{r_i} e^{-f_i(s)}, \qquad \textit{(Poisson encoding)} \quad (1)$$

where tuning curves $f_i(s)$ take the form

$$f_i(s) = \tau A \exp\left(-\tfrac{1}{2\sigma_t^2}(s - \mathring{s}_i)^2\right), \qquad \textit{(tuning curves)} \quad (2)$$

with equally-spaced preferred stimuli $\mathring{\boldsymbol{s}} = (\mathring{s}_1, \dots \mathring{s}_N)$, tuning width $\sigma_t$, amplitude $A$, and time window for counting spikes $\tau$. We assume that the tuning curves "tile", i.e., sum to a constant over the relevant stimulus range:

$$\sum_{i=1}^{N} f_i(s) = \lambda \qquad \textit{(tiling property)} \quad (3)$$

We can determine $\lambda$ by integrating the summed tuning curves (eq. 3) over the stimulus space, giving $\int ds \sum_{i=1}^{N} f_i(s) = NA\sqrt{2\pi}\sigma_t = S\lambda$, with solution:

$$\lambda = a\sigma_t/\Delta \qquad \textit{(expected total spike count)} \quad (4)$$

where $\Delta = S/N$ is the spacing between tuning curve centers, and $a = \sqrt{2\pi}A\tau$ is a constant that we will refer to as the "effective amplitude", since it depends on true tuning curve amplitude and

the time window for integrating spikes. Note, that tiling holds almost perfectly if tuning curves are broad compared to their spacing (e.g. $\sigma_t > \Delta$). However, our results hold on average for a much broader range of $\sigma_t$. (See Supplementary Figs S2 and S3 for a numerical analysis.)

Let $R = \sum r_i$ denote the total spike count from the entire population. Due to tiling, $R$ is a Poisson random variable with rate $\lambda$, regardless of the stimulus: $p(R|s) = \frac{1}{R!}\lambda^R e^{-\lambda}$.

For simplicity, we will consider stimuli drawn from a zero-mean Gaussian prior with variance $\sigma_s^2$:

$$s \sim \mathcal{N}(0, \sigma_s^2), \qquad p(s) = \frac{1}{\sqrt{2\pi}\sigma_s} e^{-\frac{s^2}{2\sigma_s^2}}. \qquad \textit{(stimulus prior)} \quad (5)$$

Since $\prod_i e^{-f_i(s)} = e^{-\lambda}$ due to the tiling assumption, the likelihood (eq. 1 as a function of $s$) and posterior both take Gaussian forms:

$$p(\boldsymbol{r}|s) \propto \prod_i f_i(s)^{r_i} \propto \mathcal{N}\left(s \,\big|\, \tfrac{1}{R}\boldsymbol{r}^\top \boldsymbol{\mathring{s}}, \tfrac{1}{R}\sigma_t^2\right) \qquad \textit{(likelihood)} \quad (6)$$

$$p(s|\boldsymbol{r}) = \mathcal{N}\left(\frac{\boldsymbol{r}^\top \boldsymbol{\mathring{s}}}{R+\rho}, \frac{\sigma_t^2}{R+\rho}\right), \qquad \textit{(posterior)} \quad (7)$$

where $\rho = \sigma_t^2/\sigma_s^2$ denotes the ratio of the tuning curve variance to prior variance. The maximum of the likelihood (eq. 6) is the so-called center-of-mass estimator estimator, $\frac{1}{R}\boldsymbol{r}^\top \boldsymbol{\mathring{s}}$, while the mean of the posteror (eq. 7) is biased toward zero by an amount that depends on $\rho$. Note that the posterior variance does not depend on which neurons emitted spikes, only the total spike count $R$, a fact that will be important for our analyses below.

## 2.1 Capacity constraints for defining optimality

Defining optimality for a population code requires some form of constraint on the capacity of the neural population, since clearly we can achieve arbitrarily narrow posteriors if we allow arbitrarily large total spike count $R$. In the following, we will consider two different biologically plausible constraints:

- A *space constraint*, in which we constrain only the number of neurons. This means that increasing the tuning width $\sigma_t$ will increase the expected population spike count $\lambda$ (see eq. 4), since more neurons will respond as tuning curves grow wider.
- An *energy constraint*, in which we fix $\lambda$ while allowing $\sigma_t$ and amplitude $A$ to vary. Here, we can make tuning curves wider but must reduce the amplitude so that total expected spike count remains fixed.

We will show that the optimal tuning depends strongly on which kind of constraint we apply.

## 2.2 Analyses based on Fisher Information

The Fisher information provides a popular, tractable metric for quantifying the efficiency of a neural code, given by $\mathbb{E}[-\frac{\partial^2}{\partial s^2} \log p(\boldsymbol{r}|s)]$, where expectation is taken with respect to encoding distribution $p(\boldsymbol{r}|s)$. For our idealized Poisson population, the total Fisher information is:

$$I_F(s) = \sum_{i=1}^N \frac{f_i'(s)^2}{f_i(s)} = \sum_{i=1}^N A\frac{(s-\mathring{s}_i)^2}{\sigma_t^4}\exp\left(-\frac{(s-\mathring{s}_i)^2}{2\sigma_t^2}\right) = \frac{a}{\sigma_t\Delta} = \frac{\lambda}{\sigma_t^2}, \quad \textit{(Fisher info)} \quad (8)$$

which we can derive, as before, using the tiling property (eq. 3). (See also Supplemental Sec. S2). The first of the two expressions at right reflects $I_F$ for the space constraint, where $\lambda$ varies implicitly as we vary $\sigma_t$. The second expresses $I_F$ under the energy constraint, where $\lambda$ is constant so that $a$ varies implicitly with $\sigma_t$. For both constraints, $I_F$ increases with increasing $a$ and decreasing $\sigma_t$ [5].

Fisher information provides a well-known bound on the variance of an unbiased estimator $\hat{s}(\boldsymbol{r})$ known as the Cramér-Rao (CR) bound, namely $\text{var}(\hat{s}|s) \geq 1/I_F(s)$. Since FI is constant over $s$ in our idealized setting, this leads to a bound on the mean squared error ([7, 12]):

$$\text{MSE} \triangleq \mathbb{E}\left[(\hat{s}(\boldsymbol{r}) - s)^2\right]_{p(\boldsymbol{r},s)} \geq \mathbb{E}\left[\frac{1}{I_F(s)}\right]_{p(s)} = \frac{\sigma_t\Delta}{a} = \frac{\sigma_t^2}{\lambda}, \qquad (9)$$

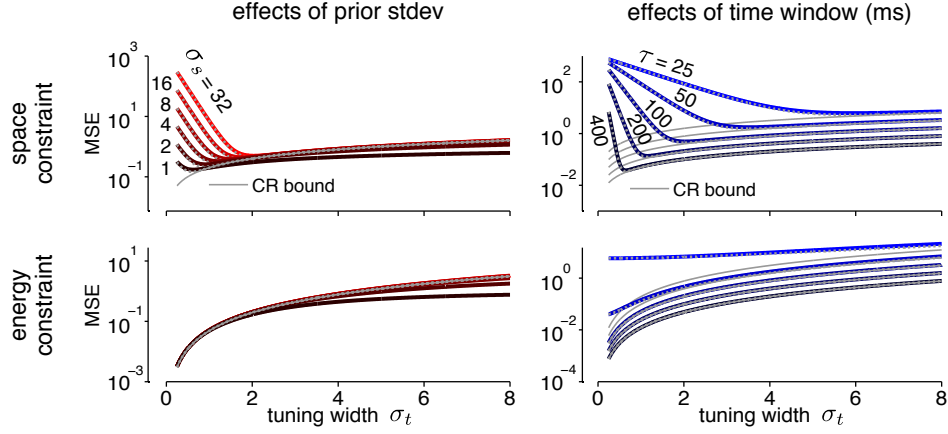

Figure 2: Mean squared error as a function of the tuning width $\sigma_t$, under space constraint (top row) and energy constraint (bottom row), for spacing $\Delta = 1$ and amplitude $A = 20$ sp/s. and **Top left:** MSE for different prior widths $\sigma_s$ (with $A$=2,$\tau = 200ms$), showing that optimal $\sigma_t$ increases with larger prior variance. Cramér-Rao bound (gray solid) is minimized at $\sigma_t = 0$, whereas bound (eq. 12, gray dashed) accurately captures shape and location of the minimum. **Top right:** Similar curves for different time windows $\tau$ for counting spikes (with $\sigma_s$=32), showing that optimal $\sigma_t$ increases for lower spike counts. **Bottom row:** Similar traces under energy constraint (where $A$ scales inversely with $\sigma_t$ so that $\lambda = \sqrt{2\pi}\tau A\sigma_t$ is constant). Although the CR bound grossly understates the true MSE for small counting windows (right), the optimal tuning is maximally narrow in this configuration, consistent with the CR curve.

which is simply the inverse of Fisher Information (eq. 8).

Fisher information also provides a (quasi) lower bound on the mutual information, since an efficient estimator (i.e., one that achieves the CR bound) has entropy upper-bounded by that of a Gaussian with variance $1/I_F$ (see [3]). In our setting this leads to the lower bound:

$$\mathrm{MI}(s, \boldsymbol{r}) \triangleq \mathrm{H}(s) - \mathrm{H}(s|\boldsymbol{r}) \geq \tfrac{1}{2} \log\left(\sigma_s^2 \frac{a}{\sigma_t \Delta}\right) = \tfrac{1}{2}\log\left(\sigma_s^2 \frac{\lambda}{\sigma_t^2}\right). \qquad (10)$$

Note that neither of these FI-based bounds apply exactly to the Bayesian setting we consider here, since Bayesian estimators are generally biased, and are inefficient in the regime of low spike counts [7]. We examine them here nonetheless (gray traces in Figs. 2 and 3) due to their prominence in the prior literature ([5, 12, 14]), and to emphasize their limitations for characterizing optimal codes.

## 2.3 Exact Bayesian analyses

In our idealized population, the total spike count $R$ is a Poisson random variable with mean $\lambda$, which allows us to compute the MSE and MI by taking expectations w.r.t. this distribution.

**Mean Squared Error (MSE)**

The mean squared error, which equals the average posterior variance (eq. 7), can be computed analytically for this model:

$$\mathrm{MSE} = \mathbb{E}\left[\frac{\sigma_t^2}{R+\rho}\right]_{p(R)} = \sum_{R=0}^{\infty} \left(\frac{\sigma_t^2}{R+\rho}\right) \frac{\lambda^R}{R!} e^{-\lambda} = \sigma_t^2\, e^{-\lambda}\, \Gamma(\rho)\, \gamma^*\left(\rho, -\lambda\right), \qquad (11)$$

where $\rho = \sigma_t^2/\sigma_s^2$ and $\gamma^*(a,z) = z^{-a}\frac{1}{\Gamma(a)}\int_0^z t^{a-1}e^{-t}dt$ is the holomorphic extension of the lower incomplete gamma function [26] (see SI for derivation). When the tuning curve is narrower than the prior (i.e., $\sigma_t^2 \leq \sigma_s^2$), we can obtain a relatively tight lower bound:

$$\mathrm{MSE} \geq \frac{\sigma_t^2}{\lambda}\left(1 - e^{-\lambda}\right) + (\sigma_s^2 - \sigma_t^2)e^{-\lambda}. \qquad (12)$$

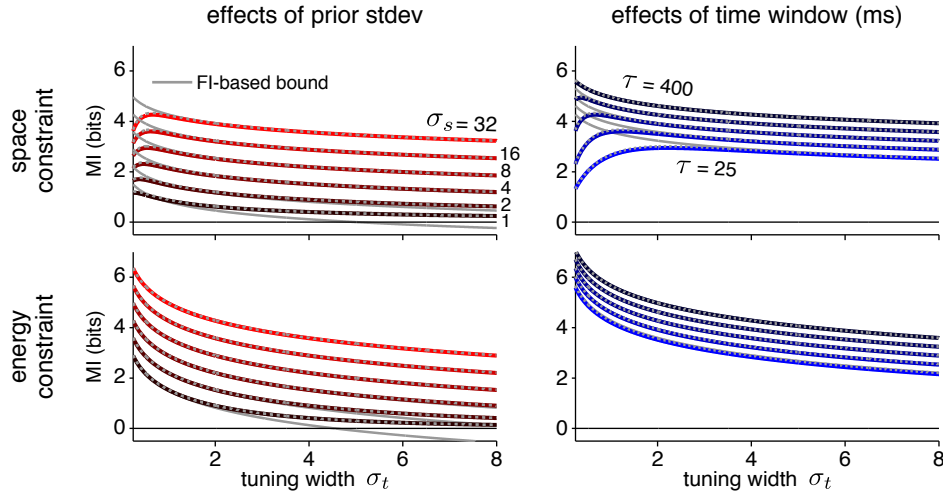

Figure 3: Mutual information as a function of tuning width $\sigma_t$, directly analogous to plots in Fig. 2. Note the problems with the lower bound on MI derived from Fisher information (top, gray traces) and the close match of the derived bound (eq. 14, dashed gray traces). The effects are similar to Fig. 2, except that MI-optimal tuning widths are slightly smaller (upper left and right) than for MSE-optimal codes. For both loss functions, optimal width is minimal under an energy constraint.

Figure 2 shows the MSE (and derived bound) as a function of the tuning width $\sigma_t$ over the range where tiling approximately holds. Note the high accuracy of the approximate formula (12, dashed gray traces) and that the FI-based bound does not actually lower-bound the MSE in the case of narrow priors (darker traces).

For the space-constrained setting (top row, obtained by substituting $\lambda = a\sigma_t/\Delta$ in eqs. 11 and 12), we observe substantial discrepancies between the true MSE and FI-based analysis. While FI suggests that optimal tuning width is near zero (down to the limits of tiling), analyses reveal that the optimal $\sigma_t$ grows with prior variance (left) and decreasing time window (right). These observations agree well with the existing literature (e.g. [15, 16]). However, if we restrict the average population firing rate (energy constraint, bottom plots), the optimal tuning curves once again approach zero. In this case, FI provides correct intuitions and better approximation of the true MSE.

**Mutual Information (MI)**

For a tiling population and Gaussian prior, mutual information between the stimulus and response is:

$$\mathrm{MI}(s, \boldsymbol{r}) = \tfrac{1}{2}\mathbb{E}\left[\log\left(1 + R\tfrac{\sigma_s^2}{\sigma_t^2}\right)\right]_{P(R)}, \tag{13}$$

which has no closed-form solution, but can be calculated efficiently with a discrete sum over $R$ from 0 to some large integer (e.g., $R = \lambda + n\sqrt{\lambda}$ to capture $n$ standard deviations above the mean). We can derive an upper bound using the Taylor expansion to log while preserving the exact zeroth order term:

$$\mathrm{MI}(s, \boldsymbol{r}) \leq \tfrac{1-e^{-\lambda}}{2}\log\left(1 + \left(\tfrac{\lambda}{1-e^{-\lambda}}\right)\tfrac{\sigma_s^2}{\sigma_t^2}\right) = \tfrac{1-e^{-a\sigma_t/\Delta}}{2}\log\left(1 + \tfrac{a}{1-e^{-a\sigma_t/\Delta}}\tfrac{\sigma_s^2}{\sigma_t\Delta}\right) \tag{14}$$

Once again, we investigate the efficiency of population coding for neurons, now in terms of the maximal MI. Figure 3 shows MI as a function of the neural tuning width $\sigma_t$. We observe a similar effect as for the MSE: the optimal tuning widths are now different from zero, but only for the space constraint. The energy constraint, as well as implications from FI indicate optimum near $\sigma_t$=0.

# 3 Poisson population coding with input noise

We can obtain a more general family of correlated population codes by considering "input noise", where the stimulus $s$ is corrupted by an additive noise $n$ (see Fig. 1):

$$s \sim \mathcal{N}(0, \sigma_s^2) \qquad \textit{(prior)} \quad (15)$$

$$n \sim \mathcal{N}(0, \sigma_n^2) \qquad \textit{(input noise)} \quad (16)$$

$$r_i | s, n \sim \text{Poiss}(f_i(s+n)) \qquad \textit{(population response)} \quad (17)$$

The use of Gaussians allows us to marginalise over $n$ analytically, resulting in a Gaussian form for the likelihood and Gaussian posterior:

$$p(\boldsymbol{r}|s) \propto \mathcal{N}\left(s \big| \tfrac{1}{R}\boldsymbol{r}^\top \boldsymbol{\check{s}}, \tfrac{1}{R}\sigma_t^2 + \sigma_n^2\right) \qquad \textit{(likelihood)} \quad (18)$$

$$p(s|\boldsymbol{r}) = \mathcal{N}\left(\frac{\boldsymbol{r}^\top \boldsymbol{\check{s}}}{\sigma_t^2/\sigma_s^2 + R(\sigma_n^2/\sigma_s^2 + 1)}, \frac{(\sigma_t^2 + R\sigma_n^2)\sigma_s^2}{\sigma_t^2 + R(\sigma_n^2 + \sigma_s^2)}\right) \qquad \textit{(posterior)} \quad (19)$$

Note that even in the limit of large spike counts, the posterior variance is non-zero, converging to $\sigma_n^2 \sigma_s^2 / (\sigma_n^2 + \sigma_s^2)$.

## 3.1 Population coding characteristics: FI, MSE, & MI

Fisher information for a population with input noise can be using the fact that the likelihood (eq. 18) is Gaussian: Eq. (18):

$$I_F(s) \triangleq -\mathbb{E}\left[\frac{d^2 \log p(\boldsymbol{r}|s)}{ds^2}\right]_{p(\boldsymbol{r}|s)} = \mathbb{E}\left[\frac{R}{\sigma_t^2 + R\sigma_n^2}\right]_{p(R)} = \frac{\lambda e^{-\lambda}}{\sigma_n^2}\Gamma(1+\rho)\gamma^*(1+\rho, -\lambda) \quad (20)$$

where $\rho = \sigma_t^2/\sigma_n^2$ and $\gamma^*(\cdot, \cdot)$ once again denotes holomorphic extension of lower incomplete gamma function. Note that for $\sigma_n = 0$, this reduces to (eq. 8).

It is straightforward to employ the results from Sec. 2.3 for the exact Bayes analyses of a Gaussian posterior (19):

$$\begin{aligned}
\text{MSE} &= \sigma_s^2 \mathbb{E}\left[\frac{\sigma_t^2 + R\sigma_n^2}{\sigma_t^2 + R(\sigma_n^2 + \sigma_s^2)}\right]_{p(R)} = \sigma_s^2 \rho\, \mathbb{E}\left[\frac{1}{\rho + R}\right]_{p(R)} + \tfrac{\sigma_s^2 \sigma_n^2}{\sigma_s^2 + \sigma_n^2}\mathbb{E}\left[\frac{R}{\rho + R}\right]_{p(R)} \\
&= \left[\rho\Gamma(\rho)\gamma^*(\rho, -\lambda) + \tfrac{\sigma_n^2}{\sigma_s^2 + \sigma_n^2}\lambda\Gamma(1+\rho)\gamma^*(1+\rho, -\lambda)\right]\sigma_s^2 e^{-\lambda},
\end{aligned} \quad (21)$$

$$\text{MI} = \tfrac{1}{2}\mathbb{E}\left[\log\left(1 + \frac{R\sigma_s^2}{\sigma_t^2 + R\sigma_n^2}\right)\right]_{p(R)}, \quad (22)$$

where $\rho = \sigma_t^2/(\sigma_s^2 + \sigma_n^2)$. Although we could not determine closed-form analytical expressions for MI, it can be computed efficiently by summing over a range of integers $[0, \ldots R_{max}]$ for which $P(R)$ has sufficient support. Note this is still a much faster procedure than estimating these values from Monte Carlo simulations.

## 3.2 Optimal tuning width under input noise

Fig. 4 shows the optimal tuning width under the space constraint: the value of $\sigma_t$ minimizing MSE (left) or maximising MI (right) as a function of the prior width $\sigma_s$, for selected time windows of integration $\tau$. Blue traces show results for a Poisson population, while green traces correspond to a population with input noise ($\sigma_n = 1$).

For both MSE and MI loss functions, optimal tuning width decreases for narrower priors. However, under input noise (green traces), the optimal tuning width saturates at the value that depends on the available number of spikes. As the prior grows wider, the growth of the optimal tuning width depends strongly on the choice of loss function: optimal $\sigma_t$ grows approximately logarithmically with $\sigma_s$ for minimizing MSE (left), but it grows much slower for maximizing MI (right). Note that for realistic prior widths (i.e. for $\sigma_s > \sigma_n$), the effects of input noise on optimal tuning width are far more substantial under MI than under MSE.

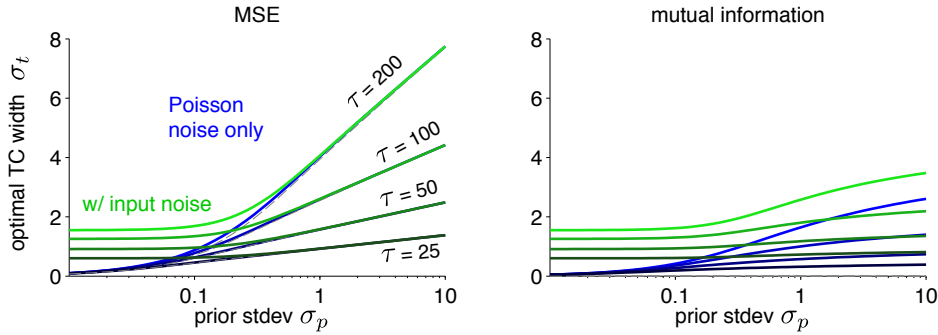

Figure 4: Optimal tuning width $\sigma_t$ (under space constraint only) as a function of prior width $\sigma_s$, for classic Poisson populations (blue) and populations with input-noise (green, $\sigma_n^2 = 1$). Different traces correspond to different time windows of integration, for $\Delta = 1$ and $A = 20$ sp/s. As $\sigma_n$ increases, the optimal tuning width increases under MI, and under MSE when $\sigma_s < \sigma_n$ (traces not shown). For MSE, predictions of the Poisson and input-noise model converge for priors $\sigma_s > \sigma_n$.

We have not shown plots for energy-constrained population codes because the optimal tuning width sits at the minimum of the range over which tiling can be said to hold, regardless of prior width, input noise level, time window, or choice of loss function. This can be seen easily in the expressions for MI (eqs. 13 and 22), in which each term in the expectation is a decreasing function of $\sigma_t$ for all $R > 0$. This suggests that, contrary to some recent arguments (e.g., [15, 16]), narrow tuning (at least down to the limit of tiling) really is best if the brain has a fixed energetic budget for spiking, as opposed to a mere constraint on the number of neurons.

## 4   Correlations induced by input noise

Input noise alters the mean, variance, and pairwise correlations of population responses in a systematic manner that we can compute directly (see Supplement for derivations). In Fig. 5 we show the effects input noise with standard deviation $\sigma_n = 0.5\Delta$, for neurons with the tuning amplitude of $A = 10$. The tuning curve (mean response) becomes slightly flatter (A), while the variance increases, especially at the flanks (B). Fig. 5C shows correlations between the two neurons with tuning curves and variance are shown in panels A-B: one pair with the same preferred orientation at zero (red) and a second with a 2 degree difference in preferred orientation (blue). From these plots, it is clear that the correlation structure depends on both the tuning as well as the stimulus. Thus, in order to describe such correlations one needs to consider the entire stimulus range, not simply the average correlation marginalized over stimuli.

Figure 5D shows the pairwise correlations across an entire population of 21 neurons given a stimulus at $s = 0$. Although we assumed Gaussian tuning curves here, one obtain similar plots for arbitrary unimodal tuning curves (see Supplement), which should make it feasible to test our predictions in real data. However, the time scale of the input noise and basic neural computations is about 10 ms. At such short spike count windows, available number of spikes is low, and so are correlations induced by input noise. With other sources of second order statistics, such as common input gains (e.g. by contrast or adaptation), these correlations might be too subtle to recover [23].

## 5   Discussion

We derived exact expressions for mean squared error and mutual information in a Bayesian analysis of: (1) an idealized Poisson population coding model; and (2) a correlated, conditionally Poisson population coding model with shared input noise. These expressions allowed us to examine the optimal tuning curve width under both loss functions, under two kinds of resource constraints. We have confirmed that optimal $\sigma_t$ diverges from predictions based on Fisher information, if the overall spike count allowed is allowed to grow with tuning width (i.e., because more neurons respond to the stimulus when tuning curves become broader). We referred to this as a "space constraint" to differentiate it from an "energy constraint", in which tuning curve amplitude scales down with tuning

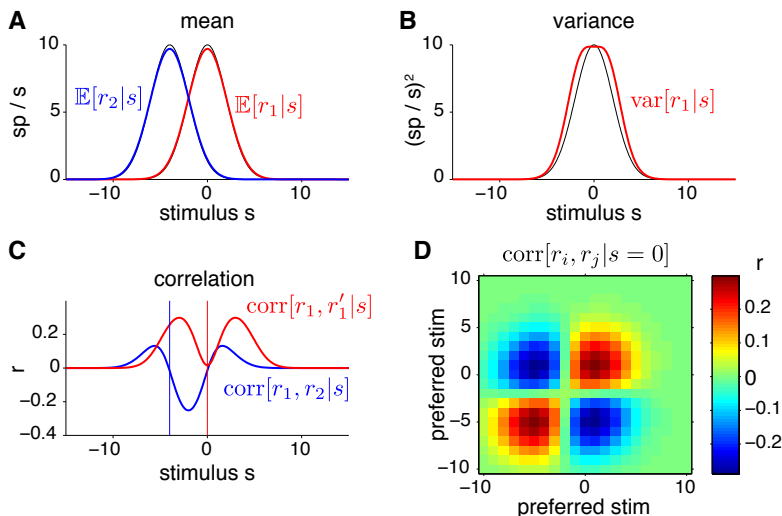

Figure 5: Response statistics of neural population with input noise, for standard deviation $\sigma_n = 0.5$. **(A)** Expected spike responses of two neurons: $\overset{*}{s}_1 = 0$ (red) and $\overset{*}{s}_2 = -2$ (blue). The common noise effectively smooths blurs the tuning curves with a Gaussian kernel of width $\sigma_n$. **(B)** Variance of neuron 1, its tuning curve replotted in black for reference. Input noise has largest influence on variance at the steepest parts of the tuning curve. **(C)** Cross-correlation of the neuron 1 with two others: one sharing the same preference (red), and one with $\overset{*}{s} = -2$ (blue). Note that correlation of two identically tuned neurons is largest at the steepest part of the tuning curve. **(D)** Spike count correlations for entire population of 21 neurons given a fixed stimulus $s = 0$, illustrating that the pattern of correlations is signal dependent.

width so that average total spike count is invariant to tuning width. In this latter scenario, predictions from Fisher information are no longer inaccurate, and we find that optimal tuning width should be narrow (down to the limit at which the tiling assumption applies), regardless of the duration, prior width, or input noise level.

We also derived explicit predictions for the dependencies (i.e., response correlations) induced by the input noise. These depend on the shape (and scale) of tuning responses, and on the amount of noise ($\sigma_n$). However, for a reasonable assumption that noise distribution is much narrower than the width of the prior (and tuning curves), under which the mean firing rate changes little, we can derive predictions for the covariances directly from the measured tuning curves. An important direction for future work will be to examine the detailed structure of correlations measured in large populations. We feel that the input noise model — which describes exactly those correlations that are most harmful for decoding — has the potential to shed light on the factors affecting the coding capacity in optimal neural populations [23].

Finally, if we return to our example from the Introduction to see how the number of neurons necessary to reach the human discrimination threshold of $\delta s$=1 degree changes in the presence of input noise. As $\sigma_n$ approaches $\delta s$, the number of neurons required goes rapidly to infinity (See Supplementary Fig. S1).

### Acknowledgments

This work was supported by the McKnight Foundation (JP), NSF CAREER Award IIS-1150186 (JP), NIMH grant MH099611 (JP) and the Gatsby Charitable Foundation (AGB).

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
