[Supplementary Material]

# Supplementary Material for

## Optimal prior-dependent neural population codes under shared input noise

Agnieszka Grabska-Barwińska & Jonathan W. Pillow
NIPS, 2014

## S1 Humans vs. Poisson population codes

Here we unpack the rough comparison of decoding performance in humans and ideal (conditionally independent) Poisson population codes provided in the Introduction.

Burr & Wijesundra 1991 [27] reports orientation discrimination thresholds ($\delta s$) as low as 0.5 deg in human observers, where threshold is defined as the angular difference at which observes achieve 81.6% correct performance in a 2AFC task.

We can relate this threshold to sensitivity (d') and Fisher information using the formula (eq. 4.4 in [10]):

$$(\delta s) \geq d'_\rho \frac{1}{\sqrt{I_F}} \tag{23}$$

where $I_F$ is the Fisher information and $d'_\rho$ is the sensitivity ($d'$) for two stimuli that can be correctly discriminated with an error probability of $\rho$, given by

$$d'_\rho = \sqrt{2}\, \Phi^{-1}(1 - \rho), \tag{24}$$

where $\Phi^{-1}(\cdot)$ is the inverse normal cumulative density function (cdf). For probability of correct $1 - \rho = 0.816$, this gives $d'_\rho \approx 1.27$. Plugging this value into (Eq. 23), it's clear that to obtain human-level discrimination performance, we need FI of at least:

$$I_f \geq (d'_\rho)^2/(ds)^2 \approx (1.27/0.5)^2 = 6.45. \tag{25}$$

Now, consider a population of 500 V1 neurons with tuning curves spaced evenly around the circle ($\Delta = 0.72$ deg), with a maximum spike rate of 50 spikes/sec, and a full bandwidth at half-height of 60 degrees (near the upper end of the range reported in monkeys [28] ). This corresponds to a Gaussian tuning width of $\sigma_t = 30\sqrt{-1/(2*\log.5)} \approx 25$ deg. This population (which clearly tiles) achieves Fisher Information (eq. 8) of approximately $I_F = \sqrt{2\pi} \times 50/(0.72 \times 25) = 7$, for all stimuli, so it is sufficient to reproduce human performance. For a population of 2000 neurons with identical characteristics, an efficient decoder could achieve a discrimination threshold twice as low as a human observer, or $(\delta s) = d'_\rho/\sqrt{I_F} \approx 0.25$ deg.

Figure S1: Number of neurons necessary to discriminate $\delta s = 1$ degree with 80% probability correct as a function of noise $\sigma_n$. We optimised Eq. 20 to match $I_F$ of the noiseless case.

## S2 Derivation of average Fisher information for tiling Poisson neurons

Fisher information for Poisson neurons and Gaussian tuning curves:

$$I_F(s) = \sum_{i=1}^{N} \frac{f'_i(s)^2}{f_i(s)} = \sum_{i=1}^{N} A \frac{(s - \mathring{s}_i)^2}{\sigma_t^4} \exp\left(-\frac{(s - \mathring{s}_i)^2}{2\sigma_t^2}\right)$$

i

The average $I_F$ per neuron equals,

$$I_F^i = \frac{1}{S} \int_{-S/2}^{S/2} I_F^i(s)ds = \frac{A}{S\sigma_t^4} \int_{-S/2}^{S/2} ds(s - \overset{\star}{s}_i)^2 \exp\left(-\frac{(s - \overset{\star}{s}_i)^2}{2\sigma_t^2}\right)$$

Assuming $S \gg \sigma_t$ (i.e. for neurons well away from the ends of the $s$-domain), we get

$$I_F^i = \frac{A}{S\sigma_t^4} \sqrt{2\pi}\sigma_t^3 = \frac{A\sqrt{2\pi}}{S\sigma_t}$$

Thus, each neuron contributes a similar average $I_F$, summing to:

$$I_F = N\frac{A\sqrt{2\pi}}{S\sigma_t} = \frac{A\sqrt{2\pi}}{\Delta\sigma_t}$$

## S3  Derivation of MSE for Poisson population code

The formula for mean squared error (MSE) in a standard Poisson population code (eq. 11) can be derived using the following series representation of the holomorphic extension of the lower incomplete Gamma function ([26], equation 8.7.1):

$$\gamma^*(a, z) = \frac{1}{\Gamma(a)} \sum_{k=0}^{\infty} \frac{(-z)^k}{k!(k + a)}. \tag{26}$$

If we substitute $\rho = (\sigma_t^2/\sigma_s^2)$ and $-\lambda$ for $a$ and $z$, respectively, then (beginning from the r.h.s. of eq. 11), we have:

$$\sigma_t^2\, e^{-\lambda}\, \Gamma(\rho)\, \gamma^*\left(\rho, -\lambda\right) = \sigma_t^2 e^{-\lambda} \sum_{R=0}^{\infty} \frac{\lambda^R}{R!(R + \rho)} = \sigma_t^2 \mathbb{E}\left[\frac{1}{R + \rho}\right]_{p(R)} = \text{MSE}, \tag{27}$$

as stated in the main text, where $p(R)$ is the Poisson distribution with mean $\lambda$.

## S4  Derivation of $I_F$ for the input noise

$$I_F = \mathbb{E}\left[\frac{R}{\sigma_t^2 + R\sigma_n^2{}^2}\right]_{p(R)} \tag{28}$$

$$= \sum_{R=0}^{\infty} \frac{R}{\sigma_t^2 + R\sigma_n^2} \frac{\lambda^R}{R!} e^{-\lambda} \tag{29}$$

$$= e^{-\lambda} \sum_{R=1}^{\infty} \frac{1}{\sigma_t^2 + R\sigma_n^2} \frac{\lambda^R}{(R - 1)!} \tag{30}$$

$$= e^{-\lambda} \sum_{R=1}^{\infty} \frac{1}{\sigma_t^2 + \sigma_n^2 + (R - 1)\sigma_n^2} \frac{\lambda^R}{(R - 1)!} \tag{31}$$

$$= \lambda e^{-\lambda} \sum_{R=0}^{\infty} \frac{1}{\sigma_t^2 + \sigma_n^2 + R\sigma_n^2} \frac{\lambda^R}{R!} \tag{32}$$

$$= \frac{1}{\sigma_n^2} \lambda e^{-\lambda} \sum_{R=0}^{\infty} \frac{1}{\sigma_t^2/\sigma_n^2 + 1 + R} \frac{\lambda^R}{R!} \tag{33}$$

$$= \frac{1}{\sigma_n^2} \lambda e^{-\lambda} \Gamma(1 + \rho) \gamma^*(1 + \rho, -\lambda) \tag{34}$$

where $\rho = \sigma_t^2/\sigma_n^2$. Combining this derivation with the above derivation for MSE (Supplement section S3) yields the terms necessary for the exact MSE under input noise (eq. 21).

## S5  Optimal tuning widths for very broad priors.

The effect of the prior width on optimal tuning width is much stronger for MSE than for MI. Empirically, we have noticed that for broad priors, the shared input noise model yields similar optimal tuning widths as the noiseless input model. From the approximation introduced in (eq. 12), we can see that the main contribution to the MSE for broad priors is $\sigma_s^2 e^{-\lambda}$. Thus, by setting $\lambda \propto 2\log\sigma_s$, we reduce the contribution of that factor to a constant. See results in Fig. S2.

Figure S2: Optimal tuning widths. Different shades of color correspond to increasing tuning amplitude, $A$. Blue depicts the optimal $\sigma_t$ for the noise-less case (as in Figs 2–3), whereas green corresponds to the input noise $\sigma_n = 1$. As we increase $\sigma_n$ (not shown here), the optimal tuning curves increase systematically for MI, and for MSE when $\sigma_s < \sigma_n$. However, for MSE, predictions of the noise-free and full model still converge for priors $\sigma_s > \sigma_n$. The dashed gray lines are optimal tuning curves obtained from the approximate lower bounds for the noiseless case (eq. 12 and 14). $\Delta=1$ degree.

## S6 Effects of imperfect tuning curve tiling

One of our main assumptions was that the tuning curves should "tile". However, the cases we considered often ventured into a range $\sigma_t \sim \Delta$, for which the tiling is not supposed to hold. We thus estimated the effect of the uneven tiling for our neuromorphic functions. We conclude that our estimates are true on average for a broad range of $\sigma_t$, reaching well below $\Delta$. That is, despite the local dependence on $s$, our metrics hold true on average.

We performed Monte Carlo simulations, whereby per each true stimulus $s$ (x-axis in Fig. 3), we simulated 1000 network responses. We estimated posteriors per each response, their MSE and entropy. In Fig. 3, 3rd and 4th column, we report the mean of these estimates as a function of $s$. One can see that even for $\sigma_t = \Delta/2$, both MSE and MI are relatively constant in $s$. However, even for the most drastic cases of $\sigma_t = \Delta/4$ and $\sigma_t = \Delta/10$, where fluctuations around the mean are clearly distinguishable, the average still matches our analytical predictions (depicted with red line in Fig. 3).

We compare the numerical and analytical predictions directly in Fig. 4. We see a robust match between the two, with small discrepancies showing only for the lowest values of $\sigma_t/\Delta$ in MI (top). In fact, discrepancies in MI most likely result from a finite sample of responses. At $A=1$, only neurons very close to the stimulus are likely to fire, with a big range of stimuli in between the peaks of tuning curves which have a much lower firing rate, and a much higher MI gain, for any of neurons that spikes for such "non-preferred" stimuli.

## S7 Correlations induced by input noise

We start by expressing changes induced by the noise in the most generic way (i.e. independent of the likelihood function). By writing response as composed of the deterministic and the noisy part: $r_i = f_i + \eta$, where by definition $\mathbb{E}\left[\eta\right] = 0$, we can write:

$$\mathbb{E}\left[r_i|s\right] = \int \frac{1}{\sqrt{2\pi}\sigma_n} e^{-\frac{n^2}{2\sigma_n^2}} f_i(s+n)\, dn \tag{35a}$$

$$\mathbb{E}\left[r_i^2|s\right] = \int \frac{1}{\sqrt{2\pi}\sigma_n} e^{-\frac{n^2}{2\sigma_n^2}} \left(f_i(s+n)^2 + \mathbb{E}\left[\eta_i^2\right]\right)\, dn \tag{35b}$$

$$\mathbb{E}\left[r_i r_j|s\right] = \int \frac{1}{\sqrt{2\pi}\sigma_n} e^{-\frac{n^2}{2\sigma_n^2}} \left(f_i(s+n)f_j(s+n) + \mathbb{E}\left[\eta_i \eta_j\right]\right)\, dn \tag{35c}$$

As we see, the mean will change in the same way regardless of the neurons' noise distribution, let us call it $\tilde{f}_i \equiv \mathbb{E}\left[r_i|s\right]$. The higher order statistics however, will depend on the noise model. For independent Poisson neurons:

$$\mathrm{var}[r_i|s] = \int f_i(s+n)^2 \frac{1}{\sqrt{2\pi}\sigma_n} e^{-\frac{n^2}{2\sigma_n^2}}\, dn + \tilde{f}_i(s) - \tilde{f}_i(s)^2 \tag{36}$$

$$\mathrm{cov}[r_i, r_j|s] = \int f_i(s+n)f_j(s+n) \frac{1}{\sqrt{2\pi}\sigma_n} e^{-\frac{n^2}{2\sigma_n^2}}\, dn - \tilde{f}_i(s)\tilde{f}_j(s) \tag{37}$$

iii

Figure S3: Analyses of imperfect tuning curve tiling. Each row shows a different value of tuning width $\sigma_t$; $A=1$. Only neurons from the middle of the coding range are shown. Average MSE (3rd column) and mutual information MI (4th column) were estimated from 1000 sample responses. The red lines in the MSE and MI columns depict our analytical predictions. Dashed lines in the fourth column demarcate mean±std (for the MSE, the standard deviation is too large to fit on the plots).

For our Gaussian tuning curves, these statistics are easy to compute, using

$$\tilde{f}(s) = \frac{A\sigma_t}{\sqrt{\sigma_n^2 + \sigma_t^2}} e^{-\frac{\left(s - \overset{*}{s}_i\right)^2}{2(\sigma_n^2 + \sigma_t^2)}}$$

and

$$\int f_i(s+n)f_j(s+n)\frac{1}{\sqrt{2\pi}\sigma_n}e^{-\frac{n^2}{2\sigma_n^2}}\,dn = \frac{(A)^2\sigma_t}{\sqrt{2\sigma_n^2+\sigma_t^2}}e^{-\frac{(\overset{*}{s}_i-\overset{*}{s}_j)^2}{4\sigma_t^2}-\frac{(s-(\overset{*}{s}_i+\overset{*}{s}_j)/2)^2}{2(\sigma_n^2+\sigma_t^2/2)}} .$$

In Fig. 5, we illustrate how neural statistics change due to input noise. For $\sigma_t = 2$ and a strong input noise of $\sigma_n = \frac{1}{4}\sigma_s$, the effects on the mean and variance are barely visible if the expected spike count is 1 (top plot). The covariance between identically tuned neurons (top right) also doesn't exceed 2% of the variance, leading to correlations not exceeding 0.05. Only for higher firing rates, the effects of noise become more visible, with covariance of identically tuned neurons reaching 20% of variance ($\rho \sim 0.4$) for $A = 10$. However, the time scale of the input noise and basic neural computations (i.e. excluding temporal integration) is about 10 ms. At such short spike count windows, available number of spikes is low.

iv

Figure S4: Comparison of exact performance measures (computed numerically by Monte Carlo simulations, 1000 samples, black pluses) with formulas derived under the assumption of a perfectly tiling tuning curves (red line), for mutual information (above, (eq. 13), and mean square error (below, (eq. 11).