[Reviews · NeurIPS 2014]

Submitted by Assigned_Reviewer_1

The authors investigate optimal population codes for one-dimensional stimulus variables under input noise. They first derive analytical expressions for the minimum mean square error and an upper bound on the mutual information for a population of conditionally independent neurons. They replicate previous numerical results showing that, unlike predicted by Fisher information, the optimal tuning width is finite and depends on the number of spikes available for decoding. They additionally show that the optimal tuning curve width also depends on the width of the prior distribution. They then generalize their expressions to the case of input noise and show that while mutual information is sensitive to input noise, the optimal tuning width with respect to mean squared error is mostly insensitive to input noise. Finally, they compute the first- and second-order statistics of the neuronal population response under input noise and briefly compare it to neural data from anesthetized monkey V1.

Overall, the paper is well written and the analytical framework is very elegant. I particularly appreciate the model description and derivation of the basic results (section 2).

Unfortunately, what seems to be the main point of the paper (sections 3 & 4) are rather short and lack intuitions. The results are merely stated (if at all) without describing the key insights that could potentially be derived from the analytical results. For instance, while the results in Eqs. 11-13 are nicely explained, such intuitions are lacking in later sections. My suggestion would be to remove section 4. Although it’s certainly a hot topic at the moment, these results aren’t directly related to optimal population coding – at least the way they are presented. The gained space could be used to expand section 3 a bit.

On a more detailed level, I have a couple of questions that struck me when reading the manuscript:

* Do Eqs. (11/12) converge for \sigma_s -> inf? In other words, is there a finite optimal tuning width if the prior is uninformative? It seems like the optimum increases as the prior gets wider, but surely that can’t be the case indefinitely?

* Why does the optimal tuning width under MSE not depend on the input noise when \sigma_n < \sigma_s, but using mutual information it does? Do the authors have some intuitions to offer?

* Are analytical solutions to Eqs. 23 & 24 available but omitted for space reasons or can these expectations not be solved in the presence of input noise? In the former case I suggest stating the results; in the latter case this would obviously weaken the power of the approach.

* While the re-parameterization in units of tuning curve spacing distance (Eq. 3) is certainly convenient and elegant, I find it hard to make sense of the numbers presented (tuning width, prior width, etc), because one always has to do mental math to convert it back to population sizes and real stimulus units.
Summary: An elegant analytical treatment of optimal population coding under input noise that formalizes earlier numerical results and generalizes them to non-uniform prior distributions.

Submitted by Assigned_Reviewer_23

Sensory systems in the brain are faced with the challenge of representing noisy signals with finite resources. Since these questions were first formalised and discussed over 50 years ago, many have pursued the optimal way of allocating resources to this task, and asked whether the brain's allocation of resources reflects this challenge. This paper attempts to further the body of literature devoted to this question. In particular, it attempts three goals: (1) to replace previous assessments of optimal coding that were based on numerical simulations with analytical results; (2) to study the implications of several components of these models (stimulus priors, the choice of objective function, the integration time, and the input noise) for optimal coding; and (3) to make predictions about how real neurons should behave under the kinds of noise considered here.

Quality: Under the (mostly reasonable) assumptions made by the authors, I believe they make some headway on the first goal, namely to analytically derive some of the previous numerical results on optimal coding. There is no discussion on how robust these results are to deviations from these assumptions. The parameter dependence is a bit of a mixed bag -- while the results on adding noise lead to generally intuitive conclusions (e.g. under noisy conditions you can't make the tuning widths arbitrarily small), the time window and objective function results I believe are already mostly known (to some extent covered across Bethge et al 2002, Brunel & Nadal 1998, Wang et al 2012, and others perhaps, eg Berens 2011 for neurometrics). The part that worries me the most though is that the third goal of this paper (presented in Section 4), seems completely tangential to the remaining work, is poorly explained, and lead to predictions that are simply not borne out by the data. The authors are able to cherry-pick two data examples which only roughly follow the predicted patterns; even in this case the predictions are a complete order of magnitude out of proportion. While the authors admit this is a weak correspondence (Line 357: "[the results] are more an exception than a rule"), the authors don't evaluate the implications of this objectively (rather, they cite an unpublished conference talk [16], as a rather weak defence).

The paper would be better if Section 4 were omitted, and the extra space devoted to better explanations of the derivations and intuitions.

Some other comments:
- The tiling property in eqn (3) is *only* an approximation, and does not hold exactly (as asserted) with finite numbers of neurons. What is the implication of the discrepancy? Does it make the decoder biased? Also this means that "the relevant stimulus range" should be defined as being well away from $\star{s}_1$ and $\star{s}_N$
- Equation (11). A reference is necessary for this equation: it took me some digging. Also some comment is necessary about the fact that this is a complex-argument extension of the incomplete gamma function, since the second argument is negative, and the $-\lambda$ factor is raised to a fractional power (which is thus complex-valued). Things always have the potential to get very messy in the complex domain (which root of $-\lambda$? do we cross a branch point?), and I can't verify that something funky hasn't happened here.
- No information was provided on how to obtain the bound in equation (12). I can't verify this.
- Also, $\sigma_{tc}$ as a symbol is a stumbling block, and would be much cleaner as $\sigma_t$.

There were several typographical errors which can be corrected.
- eqn (4): $\exp^{-\lambda}$, not $\exp^{-R}$
- Line 180: missing a factor of $2 \pi e$ inside the logarithm.
- eqn (11): exponent on $-\lambda$ should be negative. Same for equation 22.
- Fig 5 caption: s* = -5? not -2?

Clarity: While the abstract and introduction are clear, I found that the mathematics were often poorly explained. In particular, many of the results were reiterating previous findings of others (e.g. Bethge et al 2002, Berens et al 2011) that had been obtained through other analyses, but these weren't cited throughout. For example, Fig 2B seems to parallel Berens 2011 Figs 2A-B (though why is the FI different as $\sigma_{tc}$ goes to zero?). The comments on pp 185-186 are well known (and indicated as such in the intro), and this should again be stated here. I really could not follow the logic in Section 4 particularly well, not for want of trying; it didn't help that crucial information on Lines 370-372 appeared in the discussion rather than here. Also, very importantly -- there really needs to be consolidation at the end of every section/subsection otherwise this appears as an enumeration of random results.

- Line 121: what is S? The whole example is very confusing and unexplained.
- Line 124: remind that the rate is independent of s.
- Line 132: This is very unclear. Rather: "Since $\Prod_i \exp^{-f_i(s)} = \exp^{-\lambda}$ is constant, the likelihood on $s$ in (1) is thus proportional..."
- Line 134: the normal would be clearer as $\mathcal{N}( s | 1/R ... )$
- Line 146: The Fisher information, given by $FI(s) = E[ ... ] $
- Fig 5: the top line and bottom line need to be distinguished within the figure itself. The interpretations of the left and middle panels of the top line are impossible to see.
- Cramer -> Cramér

Originality: This paper does extend the previous work on this question with a new treatment. As mentioned above, it often wasn't clear what was a new contribution and what was a re-derivation or re-presentation. For those results which had previously been numerically derived, there is little comment on whether these lead to the same conclusions as the previous results obtained analytically (especially since the authors emphasise that this is a key contribution of the paper).

Significance: While the analytic treatment provides an extra degree of rigour above the previous numerical results, I am not convinced that this contribution alone is substantial. The extension to the input noise provide some results that may of use to some. However, the overall value of this paper is tainted by the lack of clear explanations or useful intuitions going along. Previous published work on this material is certainly clearer. Given that these optimality arguments have generally failed to make accurate predictions about coding in the brain in the past (with a few exceptions), and the predictions here are rather shoddy, this work convinces me more that this is a failed project than one worth pursuing. I doubt that is the authors' intention.
Summary: The paper derives some optimal coding results analytically (which is nice to see), but doesn't explain them well. The new results either are limited in scope, are not really new, or make predictions that are wrong.

Submitted by Assigned_Reviewer_27

In their paper, "Bayes optimal population codes under input noise and non-uniform priors" the authors perform an analysis of optimal population codes when neurons are correlated. The nice thing about the paper is that the authors study a scenario where it is possible to compute the MSE and the MI analytically. They use this setup to study the effect of shared input noise and nonuniform priors.

Major comments:
- In several places, the authors do not explain enough detail to follow their derivations. For example, in eq. 6, explain more details. It seems it holds because of the tiling property, but its not straightforward to see from the previous equations. In line 194, why does the derivation hold?

- In some places, I find the acknowledgement of the literature to be incomplete: The central approximation in equation 6 seems to be very similar to the work of Yaeli & Meir (2010, Front. Comp. Neuro). The observations in figure 2 agree well with the existing literature, e.g. Berens (2011, PNAS). In line 185, similar bounds have been shown by Berens et al. (2009, NIPS).

- The end of the paper is to squeezed. In particular, the analysis of the data is very superficial and the points made are not clear. I also don't agree on the interpretation of Graf et al. -> this paper does not show anything about what part of the response is due to which kind of noise. I think the paper should stop at the theoretical derivations of the correlation structure and illustrate that nicely.

Minor questions:
- many typos, just caught some ...
- Is the stimulus s bounded? Overall, I find the notation that the stimulus interval depends on N non-intuitive.
- line 48: 'For example' twice
- line 85: modeing -> l missing
- line 154: second part of the sentence is not complete
- I don't like the style "allows to easily marginalize..." if it is not really straightforward (e.g. line 256, 266, 295).
- eq. 19: what is n(s|...) supposed to mean?
- fig 4: blue should be defined in figure
- line 350: does not instead of doesn't
- line 354: missing article "the" before "available"
Summary: A very interesting and timely contribution on optimal population coding, which needs some minor adjustments
Author Feedback
Author rebuttal: We thank the reviewers for their careful reading of our paper and a thorough and constructive set of reviews. With the current set of scores, the paper seems very close to the border for acceptance, and we would therefore like to ask the reviewers for special consideration on whether to bump it up or down, in light of our replies given below.

We feel that all 3 reviewers agreed that the paper provides an interesting and potentially important extension of previous results on neural population coding, focusing on the role of input noise (a simple yet powerful way to extend population coding models to account for the finiteness of information by large populations), the prior distribution, and the role of different loss functions. However, there were also general concerns about organization, lack of intuition, and completeness of references to prior literature. We agree with these criticisms wholeheartedly, and will gladly address them fully in a revised version if given the chance. We will address these general points before turning to specific comments from each reviewer.

GENERAL:
1) Organization & clarity. All 3 reviewers suggested removing Sec 4 (a comparison of theoretical predictions and correlations in neural data), to provide more space for unpacking the results of other sections. We will gladly follow this suggestion, and agree that Sec 3 was insufficiently well explained. We will be happy to provide more intuition for these results, and have prepared a supplement with detailed derivations that which will allow others to understand and build upon these results more easily.

2) Refs to prior literature. We agree that we did not take nearly enough care crediting prior work / distinguishing which of our results are new, and we apologize profusely for this oversight. In brief, Bethge 2002 examined inadequacies of FI and indeed looked at the effects of time window and tuning curve shape; Berens 2011 examined different loss functions and effects of time window, but did not examine input noise or the effects of priors, used Gaussian as opposed to Poisson noise, and for the most part used Monte Carlo simulations instead of analytic results (due to less restrictive conditions on TCs); Yaeli & Meir 2010 also focused on decoding time, MSE, and extended to high-d stimulus variables, but did not examine MI or obtain similar analytic expressions. We will cite the relevant papers more carefully and provide a clear, thorough exposition of the results that are new in our paper.

3) Notation. Another common concern was our notation, which used tuning curve spacing (Delta) as a fundamental unit. In fact, switching back to stimulus units and writing down Delta-s explicitly turns out to yield expressions that are more comprehensible.

REVIEWER 1:

> Do Eqs. (11/12) converge for sig_s -> inf?
It turns out MSE *doesn’t* converge in this limit unless sig_tc also goes to infinity, due to a 1/R term and non-0 probability of R=0. This is interesting in light of Berens 2011, which employed a circular variable and therefore did not examine effects of stimulus range. In essence, this says we’d need infinitely many spikes to combat an infinitely broad prior distribution.

> optimal tuning under MI but not MSE depend on input noise when sig_n < sig_s?
Sorry, they differ over the entire range, but the effect is much smaller for small sig_n (eg 1% difference when sig_n = 0.1 sig_s).

> analytical solutions to Eqs. 23 & 24?
Unfortunately, analytical solutions to Eqs. (23, 24) do not exist; however, the series expressions are easy to compute, particularly given the low total spike counts. These expressions allow us to avoid Monte-Carlo simulations used previously to estimate neurometric functions, and to optimise directly.

REVIEWER #2:

We thank the reviewer for a particularly detailed set of comments, and we will make all suggested corrections that we do not have space to comment on here.

> Quality: robustness to deviations from assumptions (eg, tiling property, eq 3)
Thanks, we have examined this discrepancy carefully and found that effects are negligible until TCs become very narrow relative to their spacing (eg, when TCs are spaced 2*sig_tc apart, the “wiggles” in lambda(s) are still < 10^-3 times the mean). We will discuss this issue clearly in the revision and include exact calculations (which are still highly tractable in our setting) to illustrate robustness to this and other assumptions.

> eq (11) (complex-argument extension of the incomplete gamma function)
The reviewer is correct: we used the series expansion of the holomorphic extension of incomplete gamma. We’ll provide the full derivation in the supplement. To clarify: the term is indeed complex-valued but gives a real-valued quantity when multiplied by (-lambda)^(sigt^2/sigp^2).

> proof of bound in eq (12).
The proof relies on the fact that 1/(R+sigtc^2/sigs^2) >= 1/(R+1) when sigtc<=sigs, which allows the sum to be computed exactly.

> Line 121: what is S?
S is the stimulus range, if we were to bound it. Our intent was to show that in fact the spacing \Delta is all that matters (and not # neurons N or the range S, which can therefore be infinite).

> Line 134: normal would be clearer as $\mathcal{N}( s | 1/R ... )$
We agree, thanks for the suggestion (and for several other suggested changes related to clarity/readbility). We will also re-generate Fig. 5, as suggested.

REVIEWER 3:

We thank the reviewer for the detailed comments, and the favorable impact score.

> Is the stimulus s bounded?
No, the stimulus is technically unbounded, but in practice the results are indistinguishable if we truncate the stimulus range at several standard deviations.

> eq. 19: what is n(s|...) ?
Sorry, this denotes “a normal distribution in s with mean and variance …”

We will fix all other issues flagged by the reviewer -- thank you again.